# Mouse Sertoli Cells Inhibit Humoral-Based Immunity

**DOI:** 10.3390/ijms232112760

**Published:** 2022-10-23

**Authors:** Rachel L. Washburn, Gurvinder Kaur, Jannette M. Dufour

**Affiliations:** 1Immunology and Infectious Disease Concentration, Graduate School of Biomedical Sciences, Texas Tech University Health Sciences Center, Lubbock, TX 79424, USA; 2Department of Cell Biology and Biochemistry, Texas Tech University Health Sciences Center, Lubbock, TX 79424, USA; 3Department of Medical Education, Texas Tech University Health Sciences Center, Lubbock, TX 79424, USA

**Keywords:** Sertoli cells, complement, humoral response, transplantation

## Abstract

Transplantation is used to treat many different diseases; however, without the use of immunosuppressants, which can be toxic to the patient, grafted tissue is rejected by the immune system. Humoral immune responses, particularly antibodies and complement, are significant components in rejection. Remarkably, Sertoli cells (SCs), immunoregulatory testicular cells, survive long-term after transplantation without immunosuppression. The objective of this study was to assess SC regulation of these humoral-based immune factors. Mouse SCs survived in vitro human complement (model of robust complement-mediated rejection) and survived in vivo as allografts with little-to-no antibody or complement fragment deposition. Microarray data and ELISA analyses identified at least 14 complement inhibitory proteins expressed by mouse SCs, which inhibit complement at multiple points. Interestingly, a mouse SC line (MSC-1), which was rejected by day 20 post transplantation, also survived in vitro human complement, showed limited deposition of antibodies and complement, and expressed complement inhibitors. Together this suggests that SC inhibition of complement-mediated killing is an important component of SC immune regulation. However, other mechanisms of SC immune modulation are also likely involved in SC graft survival. Identifying the mechanisms that SCs use to achieve extended survival as allografts could be utilized to improve graft survival.

## 1. Introduction

Transplantation of human organs is a clinical treatment for various medical issues such as organ failure, cardiovascular disease, diabetes, and renal disease. Unfortunately, transplanted tissue evokes immune responses that can lead to graft rejection [1]. Subsequently, the first major hurdles to overcome in transplantation are humoral-based rejection and cell-based rejection. Humoral-based rejection occurs through serum mediators such as antibodies and complement [2]. Cell-based rejection involves activation of leukocytes and lymphocytes, particularly T cells and macrophages [2]. Due to the risk of rejection, transplant recipients are required to take chronic immunosuppressive drugs to prolong graft viability, which can cause many harsh side-effects to the patients, including frequent infections, malignancies, and organ toxicity [3].

Interestingly, Sertoli cells (SCs) have been shown to survive long-term as both allografts (donor and recipient are the same species) and xenografts (donor and recipient are different species) without immune suppressive drugs [4]. SCs are immunoregulatory cells located within the seminiferous tubules of the testis, where they function to promote spermatogenesis while protecting germ cells from immune-mediated destruction [5]. These functions are critical to male fertility [5]. In fact, understanding how SCs regulate immunity has application not only to protect germ cells in male reproduction, but also to protect grafts from immune rejection mechanism as SCs have also been shown to prolong survival of co-grafted cells [4].

An important mediator of transplant rejection is the complement system (Figure 1), which is an enzymatic cascade consisting of over 50 proteins that functions to destroy pathogens or grafted tissues [6]. Once activated in a graft, complement factors undergo a series of enzymatic cleavages, creating protein products that opsonize cells for phagocytosis, induce an inflammatory environment, and form an intermembrane pore (membrane attack complex, MAC), causing target cell lysis. The classical and alternative complement pathways are most implicated in graft rejection [7].

The classical pathway is activated by antibody binding to target cells leading to cleavage of the C4 and C2 complement components. Activation through the alternative pathway is marked by cleavage of C3 and factor B (CFB). Either way, all activation pathways converge on the assembly of the C3 and C5 convertases, which function to cleave complement components C3 and C5, respectively. The cleavage products at this stage, C3a and C5a, are anaphylatoxins that induce inflammatory responses, activate leukocytes, and recruit immune cells. The C5b fragment forms a complex with C6, C7, and C8 which binds to the target cell membrane and acts as an insertion site for C9 pore components of the MAC. The completely assembled MAC (C5b-9) now acts as an intermembrane pore to mediate cytolysis.

Furthermore, complement operates as a positive feedback loop in that, once activated, the complement response can be amplified through action of the alternative pathway. As more C3 is cleaved, it binds to more CFB, and creates an increase in opsonization proteins and C3 convertases. Multiple MAC pores are then inserted into the target cell leading to efficient lysis. Thus, to mitigate collateral damage by excessive complement activation, host cells express complement inhibitory proteins (CIPs). The various CIPs inhibit complement, acting as a kill-switch to shut down the cascade at just about every major point (Figure 1, red text).

As complement is important in graft rejection [7], the objective of this study was to analyze the effect of complement on mouse SC immune regulation. To measure survival against antibody-mediated cytolysis, we exposed mouse SCs to human serum in vitro and assessed their survival after antibody-mediated complement activation. Since this in vitro xenograft assay is a robust model of complement-mediated rejection, SC survival was especially noteworthy. To assess the SC allograft response to humoral immunity involved in acute rejection, we transplanted SCs into allogeneic mice, and then analyzed the allografts for survival, deposition of antibodies, and deposition of complement components at days 1–20 post transplantation in vivo. As primary SCs survived and showed low antibody and complement deposition, we then analyzed microarray data from mouse SCs for expression of CIPs, which are a mechanism that SC could use to protect themselves from complement-mediated destruction. ELISA assays were used to confirm protein expression of some of the lesser known but highly expressed inhibitors. Elucidating the mechanisms SCs use to survive and regulate complement could improve the survival of grafts and may even reduce the requirement of harsh, life-long immune suppressants.

## 2. Results

### 2.1. Mouse Sertoli Cells Survive Human Antibody and Complement-Mediated Killing In Vitro

To determine SC survival of antibody-activated complement, we used a human serum complement cytotoxicity assay, as xenogeneic serum has robust complement activation. Primary mouse SCs, MSC-1 cells (a mouse SC line), or control cells (porcine aortic endothelial cells, PAECs) were exposed to human AB serum containing antibodies and active complement for 1.5 h in vitro, and cell survival was assessed by MTT assay (Figure 2). Since PAEC killing by complement activation is well documented, PAECs were used as rejecting control cells [8]. While PAEC survival declined to roughly 12%, SC and MSC-1 cell survival was over 100%.

### 2.2. SC and MSC-1 Cell Allograft Survival

SCs or MSC-1 cells were transplanted as allografts underneath the kidney capsule of BALB/c mice. Graft-bearing kidneys were collected at day 20 post transplantation and immunostained for the SC marker Wilm’s tumor 1 (WT1), an SC gene important for mouse spermatogenesis that has been used previously to determine SC survival [9]. Numerous WT1-positive SCs were detected clustered together in the grafts (Figure 3A), which is consistent with previous reports using the SC marker GATA4 [10]. MSC-1 cell grafts were immunostained for large T-antigen to analyze MSC-1 cell survival. MSC-1 cells are positive for T-antigen from the SV40 virus, which was used to create this cell line [11]. By day 20 post transplantation, MSC-1 cells were completely rejected (Figure 3B) as very small grafts with lymphocytic infiltrate and no large T-antigen positive cells were detected at this timepoint. TUNEL analyses (terminal deoxynucleotidyl transferase dUTP nick end labeling assay, previously published in [10]) confirmed MSC-1 cell graft rejection. This is also consistent with our previous study demonstrating MSC-1 cells are rejected between days 11 and 20 post transplantation; thus, MSC-1 cells serve as an in vivo rejecting control graft [10]. Overall, this suggests that both SCs and MSC-1 cells may inhibit antibody and complement-mediated killing in vitro while the response in vivo is more complicated. To further analyze the importance of the humoral immune response in acute rejection, we compared antibody production, antibody binding, and complement deposition in the allografts.

### 2.3. IgG and IgM Serum Antibody Levels against SC or MSC-1 Cell Allografts

Serum from the transplanted animals (SCs or MSC-1 cells) was analyzed for production of antibodies (IgG and IgM) against SCs or MSC-1 cells. The average basal levels of IgM against SCs or MSC-1 cells in naïve BALB/c mice were 48 μg/mL, while IgG levels were below the limit of detection. No IgG production was detected throughout the study for SCs or MSC-1 cells. Mice transplanted with SCs showed no significant change in IgM response throughout the study as compared to naïve mice (Figure 4, white bars). However, a significant increase in IgM production at days 5 and 8 post transplantation was detected in mice transplanted with MSC-1 cells (Figure 4, gray bars). Furthermore, IgM levels in mice transplanted with SC grafts were significantly lower than those transplanted with MSC-1 cells at day 8 post transplantation, yet significantly higher at day 14 post transplantation.

### 2.4. Deposition of IgM and IgG on SC or MSC-1 Cell Grafts

As alloantibody binding to the surface of transplanted cells is implicated in activating complement in transplant rejection, SC or MSC-1 cell graft tissue was stained for IgM or IgG. In SC grafts, neither IgM nor IgG was detected at days 1–14 post transplantation (Table 1 and Figure 5A–F), while, at day 20 post transplantation, very little IgM and IgG deposition was detected in SC grafts (Table 1, Figure 5G,H). In MSC-1 cell grafts, IgM and IgG deposition was not detected until days 14 and 20 post transplantation (Table 1, Figure 6A–D). Again, very few cells within the MSC-1 cell grafts were positive for IgM or IgG, although elevated IgM deposition was detected as compared to IgG (Table 1, Figure 6).

### 2.5. Deposition of Complement Factors C4, C3, and MAC on SC or MSC-1 Cell Grafts

Deposition of the complement fragment C4b is indicative of the classical pathway of complement activation (Figure 1) [12]. C3b deposition on transplanted cells is indicative of complement activation and progression (Figure 1). Insertion of MAC, the terminal complex of complement lytic function, eventually results in cytolysis. Thus, to analyze complement activation, pSC or MSC-1 grafts were collected at days 1–20 post transplantation and were immunostained for C4 (Table 2, Figure 7 and Figure 8), C3 (Table 2, Figure 7 and Figure 8), and MAC (Table 2, Figure 7 and Figure 8). In both SC (Table 2, Figure 7A–H) and MSC-1 cell (Table 2, Figure 8A–H) grafts, only 1–2 cells were positive for complement factor deposition throughout the study, which is not enough to be indicative of cell death or complement activation. This suggests that both SCs and MSC-1 cells could be inhibiting the complement cascade.

### 2.6. Microarray Analyses of mRNA Complement Inhibitor Expression in SC and MSC-1 Cells

One mechanism SCs may use to inhibit complement would be expression of complement inhibitors. CIPs act as kill-switches to shut down the complement cascade at almost every step (Figure 1, red text) to mitigate collateral damage to host cells by excessive complement activation. Thus, analysis of the CIPs expressed by SC and MSC-1 cells was conducted using our previously generated microarray data [10]. At least 14 CIPs were found to be expressed by SCs (Table 3). Nine of these inhibitors have never been identified to be expressed by SCs. Of these inhibitors, SCs express C1INH, COMP, Factor I, CD55, clusterin, CPB, and CPN significantly higher than MSC-1 cells, and MSC-1 cells express C1qI and CD59 significantly higher than SCs (*p* < 0.05). Previously, we confirmed expression of the complement inhibitors C1INH, CD46, CD55, CD59, and clusterin by qPCR [10]. Together, expression of all these identified inhibitors has the potential to inhibit all major parts of the complement cascade including activation, amplification, and lysis, and this should be investigated in future studies. In spite of inhibiting complement, MSC-1 cells are still completely rejected by day 20 post transplantation, indicating that complement is not a major player in the acute rejection of MSC-1 cells, and that they are being rejected by another mechanism.

### 2.7. Protein Expression of C1INH, C1qI, and COMP Inhibitors by SC

Previously, our group and other groups confirmed protein expression for the CIPs CD46, CD55, CD59, clusterin, and Crry in mouse, rat, or pig SC [10,13,14,15,16,17]. Thus, we performed ELISA assays to quantify protein expression of the highly expressed but less studied inhibitors C1INH, C1qI, and COMP. As these are all secreted factors, we analyzed Sertoli cell conditioned medium (SCCM) to determine if mouse SC express and secrete these inhibitors (Figure 9). We confirmed that mouse SC express protein for C1INH at 133.065 ± 9.001 ng/mL, C1qI at 1.205 ± 0.092 ng/mL, and COMP at 11.560 ± 0.231 ng/mL.

## 3. Discussion

In this study, we sought to understand the mechanisms via which SCs survive antibody-activation of the complement system. Therefore, we used both an in vitro xenogeneic hyperacute assay and an in vivo allotransplantation model. Antibody-mediated activation of the complement cascade is an important mediator of hyperacute, acute, and chronic graft rejection. In fact, antibody-mediated rejection is the most common cause of late graft failure [18]. Further study of tissues, such as immunoregulatory SCs, that survive when transplanted as allografts or xenografts independent of immunosuppressants could lead to improved treatments to create a graft-protective environment. Moreover, autoimmune orchitis involves inflammation and antibodies to spermatogenic germ cells. The presence of anti-sperm antibodies is associated with male infertility. Further understanding of how SCs could regulate the humoral immune response is also relevant to prevention of autoimmune orchitis.

Here, we first examined SC survival after exposure to human serum with complement and antibodies. This assay was used to create a robust hyperacute killing environment with both classical and alternative pathway activation of complement-mediated cell lysis in vitro. The robust killing associated with this assay was confirmed by nearly 90% cell death of the PAECs. In contrast, both SCs and MSC-1 cells survived with over 100% viability in the human serum. This increase in survival may be due to the presence of over 50% serum, which contains growth factors that may stimulate SC proliferation. These results corroborate our previous published findings, where we reported that pig SCs survived human serum with rabbit complement, while pig islets and PAECs were killed [8,16]. Surprisingly, MSC-1 cells, which we have shown are rejected in vivo when transplanted as allografts, demonstrated similar survival as SCs, indicating they may maintain similar complement-modulatory properties as SCs.

In vivo survival of SC and MSC-1 cell allografts was assessed, and grafts were analyzed for antibody activation of the complement system. While SC grafts survived through day 20 post transplantation in 100% of the mice, MSC-1 cells were completely rejected by day 20 post transplantation. This is consistent with previous studies where allogeneic and xenogeneic SCs were present for over 3 months after transplantation [19,20], while MSC-1 cells were found to start rejecting by day 8 post transplantation and were no longer present by day 20 [10]—the typical allograft rejection period [21]. One concern for long-term graft survival is tumor formation. However, despite the expression of WT1, SCs in the testis are considered terminally differentiated cells and very rarely form tumors. Previous analysis of SC grafts for proliferation markers (proliferating cell nuclear antigen and Ki67) or BrdU labeling found that transplanted SCs stop proliferating within the first 14 days after transplantation and do not form tumors [22].

IgM production was significantly higher in animals transplanted with MSC-1 cells at days 5 and 8 post transplantation as compared to pre-transplantation values, suggesting that the humoral immune response was activated against these cells. No significant change in sera IgM was seen in SC grafts as compared to non-transplanted mice. IgM immunoglobulins are the first antibodies generated in the initial immune response. IgM is also the most effective type of antibody to activate complement; thus, an increase in sera IgM in MSC-1 cell-transplanted animals, which eventually rejected their grafts, indicates active humoral immunity [23].

IgG is the only other immunoglobulin class that can activate complement, albeit less efficiently than IgM. IgG antibodies specific for transplanted tissue would begin to be produced after immune recognition of the graft, and the levels would rise over time, sometimes taking days to weeks, until the immune response is concluded. An antigen-specific IgG response was not detected in either set of graft recipients. It is possible that SCs could be inhibiting antibody production as cell culture experiments have demonstrated that SC conditioned medium can inhibit the proliferation of B cells in vitro [24,25,26].

Even though serum IgG production specific to SCs and MSC-1 cells was not detected in either SC or MSC-1 cell-transplanted animals by ELISA, IgG deposition was detected by immunohistochemistry at later timepoints in SC or MSC-1 cell grafts. This discrepancy could be explained by the location of the deposited IgG antibody in the grafts, as antibodies were not bound to SC. The IgG deposition was detected mainly on the cellular infiltrate (small, round immune cells) or cellular debris in SC or MSC-1 cell grafts, respectively. Thus, it is possible that bound IgG is not specific to SCs or MSC-1 cells, which would explain why we were unable to detect IgG specific to SCs or MSC-1 cells at later timepoints by ELISA.

Antibody deposition (both IgG and IgM) was not detected in SC or MSC-1 grafts at early timepoints and suggests that both SCs and MSC-1 cells could be inhibiting the binding of alloantibodies, but the mechanism for this inhibition needs further investigation. Antibody deposition at later timepoints (days 14 and 20 post transplantation) on MSC-1 cell grafts could be mediating opsonization to clean up cell debris as most of the MSC-1 cells were already rejected at this timepoint [27]. In SC grafts, the antibody deposition at day 20 could be involved in activating the complement cascade to destroy SCs by chronic rejection. However, complement deposition (C4, C3, and MAC) was not detected at this timepoint, which implies that SCs also inhibit complement-mediated cytolysis at later timepoints. Furthermore, it has previously been shown that SCs (mouse and pig) survived over 90 days post transplantation, indicating that SCs are surviving chronic rejection mediated by antibody deposition, as well as other means of graft rejection [8,28].

Complement activation can also occur by the alternative pathway, which does not require the deposition of antibodies and accounts for roughly 30% of the cell lysis of pig islets [16,19]. Thus, to assess the role of complement cascade activation through the alternative pathway (earlier timepoints, no antibody deposition) and the classical pathway (later timepoints, days 14 and 20 post transplantation) SC and MSC-1 cell grafts were analyzed for deposition of the complement factors C4 (classical pathway activation), C3 (alternative pathway activation and amplification), and MAC (terminal pathway and cell lysis). No significant complement factor deposition was detected in SC or MSC-1 cell grafts throughout the study suggesting that, similar to the in vitro data, SCs are capable of inhibiting both the alternative and the classical pathways of complement-mediated cell lysis. These results are consistent with our previously published xenograft data where we demonstrated that pig SCs actively inhibit both the alternative and the classical pathways of complement-mediated cell lysis, while the alternative pathway is involved in rejecting pig islets in vitro and in vivo [8,16].

Even though IgM antibodies specific to SCs and MSC-1 cells were detected in serum from naïve and transplanted mice, our data suggest that both SCs and MSC-1 cells are capable of inhibiting complement immediately after transplantation (days 1–2 post transplantation) and during the acute (days 14–20 post transplantation) graft rejection periods. Interestingly, MSC-1 cells, which lack many of the immune-privileged properties associated with SCs, are also able to inhibit complement. However, the eventual rejection of MSC-1 cells suggests that the activated IgM response could have primed the cell-mediated immune responses, such as macrophages and T cells, which play primary roles in acute rejection.

Since MSC-1 cells survived complement-mediated cell lysis in vitro but were still rejected after transplantation as allografts, we performed microarray analyses on the SCs and MSC-1 cells and compared the results to identify CIPs that may be critical for escaping the immune response [10]. We identified mRNA expression of at least 14 CIPs, namely, C1INH, C1qI, PTX3, COMP, SUSD4, Crry, CFH, CFI, CD46, CD55, CD59, clusterin, CPB, and CPN, by SCs, with both SCs and MSC-1 cells expressing seven different inhibitors which may be increasing their survival. Western blots have previously been performed to confirm protein expression of CD46, CD55, CD59, clusterin, and Crry in SCs [10,13,14,15,16]. Using ELISA assays, we confirmed SC protein expression of the secreted inhibitors with the highest RNA expression: C1INH, C1qI, and COMP (Figure 9).

A significant increase in expression of seven CIPs was measured in SCs versus MSC-1 cells including C1INH, COMP, CD55, clusterin, and the carboxypeptidases, which inhibit the C1 complex of the classical pathway, MASP-1/-2 of the lectin pathway, the C3 and C5 convertases of the amplification loop, MAC, and anaphylatoxins. Conversely, MSC-1 cells had significantly elevated expression of two CIPs: C1qI, which inhibits the C1 complex of the classical pathway, and CD59, which inhibits MAC assembly and insertion. These significant differences in expression may explain in part why SCs are not rejected in immune competent mice when MSC-1 cells are rejected; they are better equipped to decrease activity at every major part of the complement cascade from activation to termination, amplification, inflammation, opsonization, and cytolysis. This suggest that, even though MSC-1 cells survive the in vitro complement assay, additional factors are necessary to survive the combined humoral and cell-mediated immune rejection response.

Recently, clinical xenotransplantation of transgenic pig kidneys (into beating-heart cadavers) [29] and a pig heart [30] was performed in the United States. Of great significance is the cardiac xenotransplantation of a genetically modified pig heart into a living patient to treat life-threatening heart failure [30]. The pig heart had 10 gene modifications including knockout of three carbohydrate xenoantigens (to decrease preformed antibody binding), knockout of growth hormone receptor (to prevent cardiac growth), knock-in of CD46 and CD55 (to inhibit complement), endothelial cell protein C receptor, and thrombomodulin (thromboregulatory proteins), and knock-in of heme oxygenase 1 and CD47 (to reduce inflammatory response) [30]. The recipient was also required to undergo extensive immunosuppression, which involved traditional immune suppressants and a novel CD40 co-stimulation blockade drug. The patient survived 7 weeks before the heart failed [30]. An endomyocardial biopsy was performed on day 50 post transplantation, and both IgG and IgM were present on damaged capillary tissue and in serum, but C3d and C4d were not detected by immunostaining [30]. This biopsy was repeated 6 days later and showed positive C4d staining (particularly on the necrotic tissue) and a decrease in IgG and IgM staining on capillary tissue [30]. Thus, an increase in antibody binding and production was observed, but complement-mediated killing was not. Even though the organs eventually failed, these ground-breaking clinical studies suggest that xenotransplantation is possible. Understanding of rejection and survival mechanisms in transplantation continue to increase, demonstrating the possibility to further uncover ways to improve graft survival, which may be beneficial for transplant recipients. As SCs survive long-term as allografts and xenografts, and since they express at least 14 CIPs (12 more than the genetically modified pigs), these additional factors may provide clues that could be useful in this transgenic pig model to continue to increase its impressive viability.

Overall, our in vitro data demonstrate that SCs and MSC-1 cells survive human complement, and our in vivo data demonstrate that SCs and MSC-1 cells inhibit complement deposition. SC grafts have been shown to survive over 100 days post transplantation, but MSC-1 cell grafts are rejected within 20 days, even though both cell types inhibit complement. MSC-1 cell rejection could be due to cell-mediated apoptosis, which is consistent with our previous TUNEL assay findings [10]. These results suggest that even though the humoral immune response is an important part of graft rejection, and that SC inhibition of this response is necessary for their survival, they must be regulating other parts of the immune response, such as the cell-mediated response, in order to survive. Understanding how SCs establish a graft-protective environment could aid in overcoming the major hurdles of transplant rejection and toxic immune suppression, which would increase access of clinical transplantation for patients.

## 4. Materials and Methods

### 4.1. Animals

Testes were collected from male C57BL/6x 129 mice (The Jackson Laboratory, Bar Harbor, ME, USA), 19–20 days old for SC isolation. Male BALB/c mice (Charles River Laboratories, Wilmington, MA, USA) 6–8 weeks of age were used as allogeneic recipients. All animals were housed and maintained at appropriate conditions in adherence to the approved Institute for Laboratory Animal Research Care, Use of Laboratory Animals, Texas Tech University Health Sciences Center Institutional Animal Care and Use Committee guidelines and protocols of the National Institutes of Health.

### 4.2. Cell Preparation and Transplantation

SCs were isolated from testes using collagenase and trypsin/DNase digestion, as described previously [10,31]. Briefly, after testes were thoroughly chopped in a sterile hood, tissue was washed three times in Hank’s balanced salt solution (HBSS) and centrifuged to remove red blood cells. Then, a collagenase digest was performed to break up seminiferous tubules, followed by HBSS washes. Next, the suspension was incubated while shaking in dissociation medium, trypsin, and DNase to dissociate cells. This was followed by one wash after which the cell suspension was placed through a filter to separate SC from suspension. The SCs were cultured as aggregates in non-tissue culture plates using supplemented Ham’s F10 medium for 48 h at 37 °C before transplantation [10,31]. The MSC-1 cell line was derived from a SC tumor collected from testes in transgenic C57BL/6x SJL mixed hybrid mice. These mice carried a transgene containing DNA encoding both the small and the large T antigen of the SV40 virus fused to the promoter for human Mullerian-inhibiting substance [28]. Since Mullerian-inhibiting substance is specific to SCs in males, this led to a SC tumor specific for T antigen. MSC-1 cells have some, but not all, of the immunoregulatory properties of SCs. Prior to transplantation, MSC-1 cells were also cultured and aggregated in the same manner as the SCs. Four million MSC-1 or SCs (estimated using a DNA assay [10,28,31]) were transplanted into the left renal subcapsular space of naïve isoflurane anesthetized BALB/c mice [10,28,31].

### 4.3. Human Serum Cytotoxicity Assay

The in vitro human serum cytotoxicity assays were performed similarly to that described previously [8]. This xenogeneic in vitro assay was used to analyze SC survival to antibody-activated complement because this is a model of robust complement-mediated killing in hyperacute rejection, and survival during this assay would indicate inhibition of complement. Briefly, 2 × 10^5^ SCs (*n* = 3), 2 × 10^5^ MSC-1 cells (*n* = 3), or 1 × 10^5^ PAECs (*n* = 3) were plated per well on a 24-well tissue culture plate (Becton Dickinson Labware, Franklin Lakes, NJ, USA) and cultured overnight in supplemented Dulbecco’s modified Eagle medium (DMEM) + 10% fetal bovine serum (FBS). The next morning, half of the medium was removed per well, and the cells were incubated at 37 °C for 1.5 h in medium only or in 50% medium and 50% pooled AB human serum containing antibodies and complement (Innovative Research, Inc, Novi, MI, USA). At the end of the incubation, medium was removed and cell survival was measured using the Cell Proliferation Kit MTT (3-[4,5-deimethylthiazol-2-yl]-2,5-deiphenyltetrazolium bromide) assay (Millipore-Sigma, Darmstadt, Germany) as described previously [8].

### 4.4. ELISA for IgG and IgM

Blood from BALB/c mice transplanted with SCs or MSC-1 cells was collected at days 1, 2, 5, 8, 11, 14, and 20 post transplantation (*n* ≥ 3). Blood was also collected from naïve, non-transplanted mice to measure basal IgG and IgM levels. Serum was separated from red blood cells by centrifugation (1200× *g* for 10 min) and stored at −80 °C until analyzed. Sera were screened for IgG and IgM antibodies against SCs and MSC-1 cells via indirect ELISA (Bethyl Laboratories Inc., Montgomery, TX, USA). SCs or MSC-1 cells (10,000 per well) were plated on a 96-well tissue culture plate in DMEM + 10% FCS for 24 h at 37 °C. The cells were fixed with 4% paraformaldehyde for 10 min and used as antigen for indirect ELISA. Nonspecific binding was blocked by addition of 50 mM Tris, 0.14 M NaCl, and 1% bovine serum albumin (BSA) followed by incubation with horseradish peroxidase (HRP)-labeled goat anti-mouse IgG (1:40,000 Bethyl Laboratories) or IgM (1:50,000; Bethyl Laboratories) reagents. After the final wash, 3,3′,5,5′-tetramethylbenzidine peroxidase substrate was added to develop the reaction. The reaction was terminated after roughly 15 min via addition of 2 M H_2_SO_4_, and plates were read at an optical density (OD) of 450 nm. OD values of samples were converted to micrograms per milliliter (μg/mL) using standard curves generated with mouse reference serum (0.0078–5 μg/mL for IgG and 0.015625–1 μg/mL for IgM). The final concentration was obtained by multiplying by the dilution factor.

### 4.5. Graft Characterization

SC or MSC-1 cell graft-bearing kidneys were collected at days 1–20 post transplantation, and tissue sections were paraffin-embedded, processed, and immunostained. To identify antibody deposition, goat anti-mouse IgG (1:450, Bethyl Laboratories, Montgomery, TX, USA) and IgM (1:250, Bethyl Laboratories) antibodies were used. Tissue sections were further analyzed for complement factor deposition, specifically C4 (polyclonal anti-human C4; 1:2000; Calbiochem, Darmstadt, Germany), C3 (polyclonal anti-human C3; 1:4000; Calbiochem), and membrane attack complex (polyclonal anti-human MAC; 1:200; Calbiochem). After incubation with these antibodies, sections were incubated with biotinylated secondary antibodies (1:200; Vector Laboratories, Burlingame, CA, USA) followed by the ABC-enzyme complex (Vector Laboratories) with diaminobenzidine (Biogenex Laboratories) as a chromagen. To detect cell nuclei, all sections were counterstained with hematoxylin. Negative controls put through the same procedure without primary antibody lacked a positive signal.

### 4.6. Bioinformatic Analyses

After aggregating in culture for 2 days at 37 °C and 5% CO_2_, SCs (*n* = 3) or MSC-1 cells (*n* = 3) were lysed using 1 mL of Trizol reagent. RNA extraction was performed using the protocol from the manufacturer (Invitrogen Corp., Carlsbad, CA, USA). RNA quality was determined by formaldehyde agarose gel electrophoresis as described in Doyle et al. (2012) [10]. Mouse Expression 430 2.0 microarrays with 45,101 total probes and 23,843 genes (Affymetrix, Santa Clara, CA, USA) were used for transcriptome profiling [10]. Microarray processing and data analyses were performed by Doyle et al. (2012) [10]. Briefly, the cDNA was used to create antisense biotin-labeled cRNA, which was then fragmented and hybridized using GeneChip probe array. To determine raw signal intensity, cRNA was incubated with streptavidin–phycoerythrin conjugate, and image files were detected and analyzed with Affymetrix Genechip model 3000 scanner and Operating Software (Affymetrix, Santa Clara, CA, USA). The data were normalized to at least one sample with a raw signal of ≥50 and both samples with a normalized signal of ≥0.025 to get rid of excess noise. ANOVA was performed with a *p*-value of ≤0.05, and significant probes were chosen with a log-fold change ±4.0 of SC as compared to MSC-1 [10]. Microarray data were previously analyzed for immunoprotective properties (including cytokines such as TGF-β, IL-6, and IL-1) of SCs as compared to MSC-1 cells in Doyle et al. (2012) [10]. Here, we analyzed the microarray data under these filtration parameters for the expression of complement inhibitory proteins.

### 4.7. ELISA for Secreted Complement Inhibitors

About 15 million mouse SCs (*n* = 3) were cultured overnight on 150 mm tissue culture plates (Corning Inc., Corning, NY, USA) in 35 mL of DMEM + 10% FBS. Sertoli cell conditioned medium (SCCM) was collected and assayed per the manufacturer’s protocol by ELISA (Aviva Systems Biology, San Diego, CA, USA) for C1INH (SERPING1), C1qI, or COMP. Briefly, standards or diluted samples (1:10 dilution) were added to a precoated well and were incubated at 37 °C for 1–2 h depending on which inhibitor was being assayed (per manufacturer’s instructions). Wells were then incubated with the provided biotinylated detector antibody for 60 min. Wells were washed with wash buffer followed by incubation with avidin–horse radish peroxidase (HRP) for 30–60 min. Lastly, the 3,3′,5,5′-tetramethylbenzidine (TMB) substrate was added per well, and plates were incubated for 20–30 min (depending on ELISA kit), followed by stop solution. Plates were read within 5 min at an OD absorbance of 450 nm.

### 4.8. Statistical Analysis

All values are expressed as the means ± standard error of mean and were compared using one-way ANOVA or multiple *t*-tests. Statistical significance between groups was set at a *p*-value < 0.05. All statistical analyses were performed using GraphPad InStat and Prism software.

## 5. Conclusions

Overall, these data demonstrate that mouse SCs survive xenogeneic human antibody and complement-mediated killing in vitro, as well as allogeneic complement-mediated destruction without immune suppressants in mice, and that SCs express most of the known complement inhibitors. Since MSC-1 cells also expressed CIPs and survived the in vitro human serum assay but were rejected after allograft transplantation, this indicates that SC survival of acute rejection is more complicated than solely inhibiting complement-mediated cytolysis, and that other mechanisms may be at play including regulation of cellular immunity. Thus, investigation of SC modulation of cellular rejection mechanisms should be investigated in the future. Additionally, complement inhibition may play a larger role in decreasing immune cell infiltration during acute rejection and in protecting allografts against chronic rejection, which is marked by ischemia, vascular occlusion, and organ death months to years after transplantation. Elucidating how SCs achieve this graft survival will further knowledge in graft protection with a decreased requirement of toxic immune suppressant drugs.

## Figures and Tables

**Figure 1 ijms-23-12760-f001:**
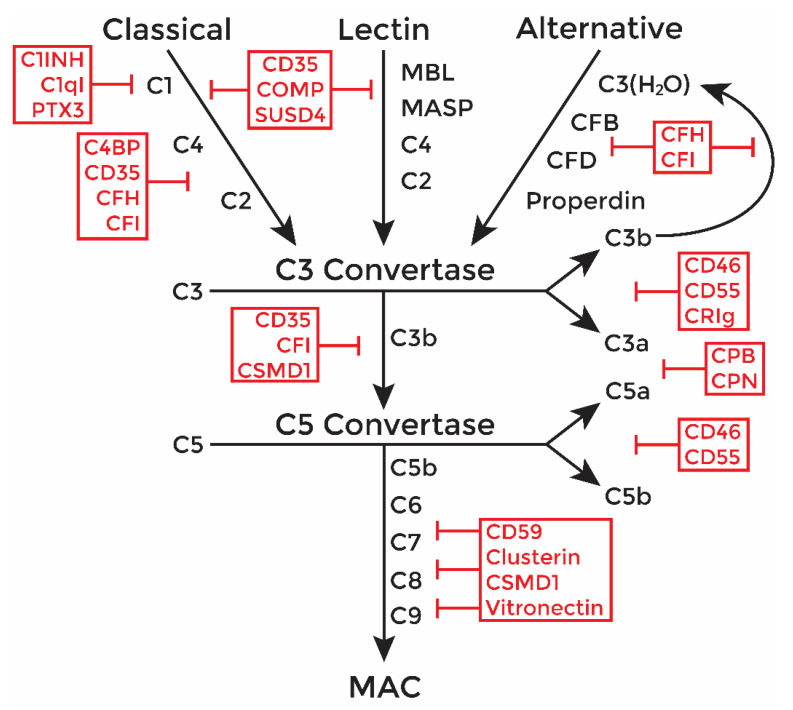
The complement system. Complement can be activated through the classical (by antibody binding), lectin (by bacterial oligosaccharides and acetylated residues), or alternative (spontaneously or by other pathway activation) pathways. All three pathways converge at the formation of a C3 convertase, which then forms the C5 convertase, which cleave C3 and C5 into their respective anaphylatoxins (C3a, C5a) and their pathway components (C3b, C5b). Complement terminates with assembly and insertion of the membrane attack complex (MAC), an intermembrane pore that causes cytolysis of the target cell. Complement cascade components are in black, and inhibitors are in red. C1INH: C1 inhibitor (SERPING1). C1qI: C1q inhibitor. C4BP: C4 binding protein. CD35: complement receptor 1 (CR1). CD46: membrane cofactor protein (MCP). CD55: decay accelerating factor (DAF). CFH: complement factor H. CFI: complement factor I. COMP: cartilage oligomeric matrix protein. CPB: carboxypeptidase B. CPN: carboxypeptidase N. CRIg: complement receptor of the immunoglobulin superfamily. CSMD1: CUB and sushi domain protein 1. PTX3: pentraxin 3. SUSD4: sushi domain-containing protein 4.

**Figure 2 ijms-23-12760-f002:**
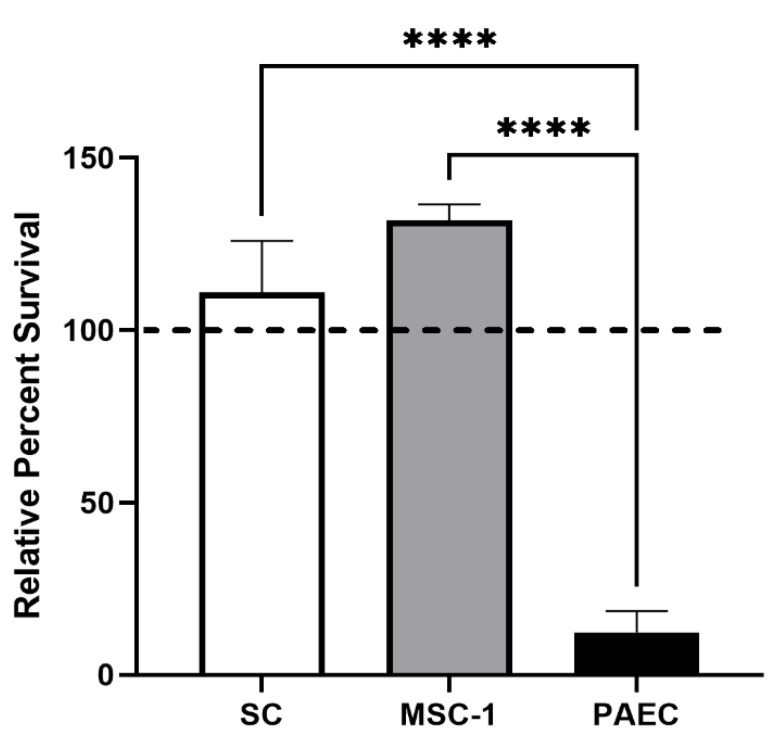
Mouse SCs survive after exposure to human serum in vitro. SCs (white bar), MSC-1 cells (gray bar), and control cells (PAECs, black bar) were exposed to human AB serum containing antibodies and active complement. Cell survival was assessed by MTT cell viability assay. Cell viability for cells cultured in medium only were set to 100% (dashed line) and the relative percentage viability was calculated for each cell type. Viability is presented as the mean ± SEM for at least three different experiments. The significance was determined by ANOVA and Fisher’s PLSD. **** indicates *p* < 0.0001.

**Figure 3 ijms-23-12760-f003:**
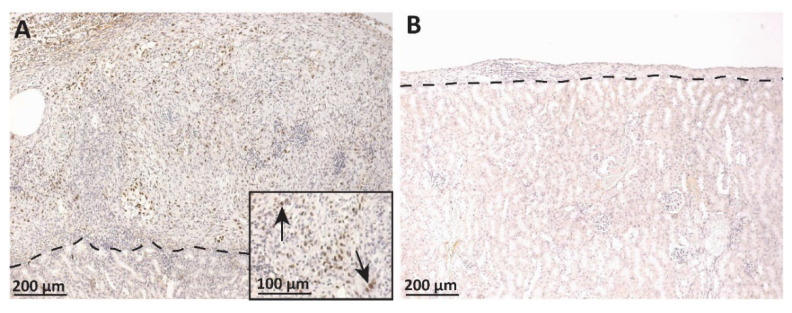
Immunohistochemical analysis of SC or MSC-1 cell allograft survival. Four million SCs or MSC-1 cells were transplanted under the kidney capsule of BALB/c mice. The grafts were collected on day 20 post transplantation and immunostained for SCs (WT1, brown (**A**) or MSC-1 cells, large T-antigen, brown (**B**)). Inset in (**A**) is a higher magnification and is included for visualization of WT1 positive cells. A dotted line separates the graft (above the line) from the kidney (below line). Sections were counterstained with hematoxylin (blue). Arrows indicate examples of positively stained cells.

**Figure 4 ijms-23-12760-f004:**
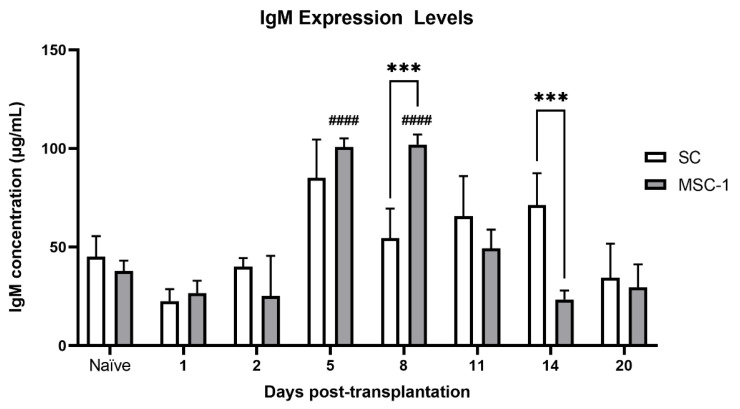
Production of IgM antibodies to grafted SCs or MSC-1 cells. Serum from BALB/c mice transplanted with SCs or MSC-1 cells was collected at days 1, 2, 5, 8, 11, 14, and 20 post transplantation. Serum was also collected from naïve BALB/c mice (mice that did not receive transplanted cells). The production of IgM (μg/mL) against SC (white bar) or MSC-1 (gray bar) grafts was graphed. Data shown are the mean ± SEM for at least three different experiments per timepoint. *** denotes a significant difference of the means as determined by multiple *t*-tests at *p* ≤ 0.001 as comparing SC and MSC-1 levels against each other; ^####^ denotes a significant difference of the means determined by one-way ANOVA at *p* ≤ 0.0001 as compared to respective naïve, pre-transplantation values.

**Figure 5 ijms-23-12760-f005:**
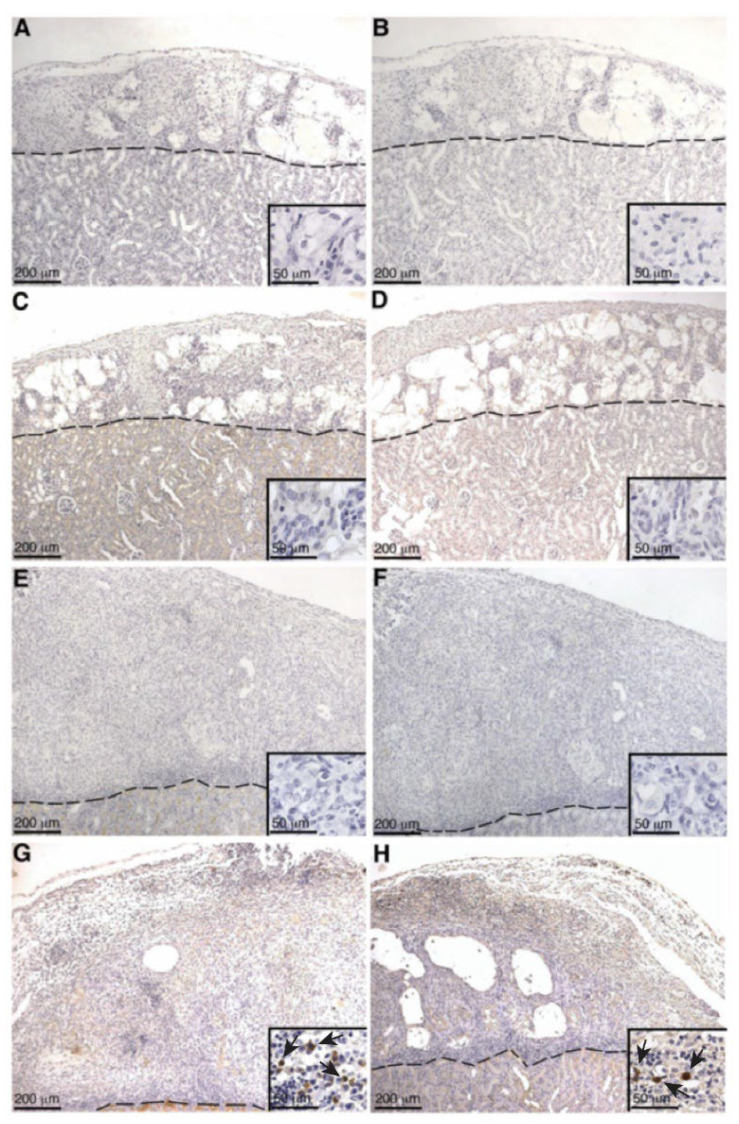
Deposition of antibodies (IgG or IgM) on SC grafts. Tissue sections were collected at days 2 (**A**,**B**), 5 (**C**,**D**), 14 (**E**,**F**), and 20 (**G**,**H**) post transplantation and immunostained for IgG (brown; **A**, **C**,**E**,**G**) or IgM (brown; **B**,**D**,**F**,**H**). Insets are higher magnifications. A dotted line separates the graft (above the line) from the kidney (below the line). Hematoxylin was used to stain the cell nuclei (blue). Arrows indicate examples of positive staining.

**Figure 6 ijms-23-12760-f006:**
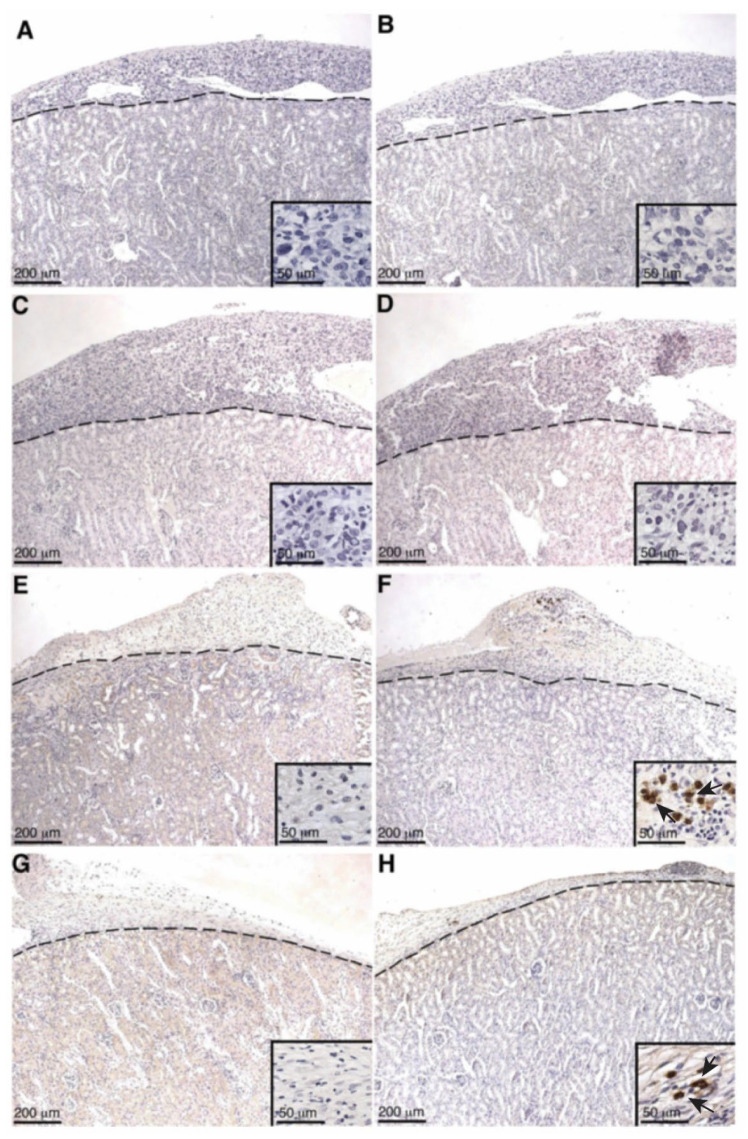
Deposition of antibodies (IgG or IgM) on MSC-1 grafts. Tissue sections were collected at days 2 (**A**,**B**), 5 (**C**,**D**), 14 (**E**,**F**), and 20 (**G**,**H**) post transplantation and immunostained for IgG (brown; **A**,**C**,**E**,**G**) or IgM (brown; **B**,**D**,**F**,**H**). Insets are higher magnifications. A dotted line separates the graft (above the line) from the kidney (below the line). Sections were counterstained with hematoxylin (blue). Arrows indicate examples of positive staining.

**Figure 7 ijms-23-12760-f007:**
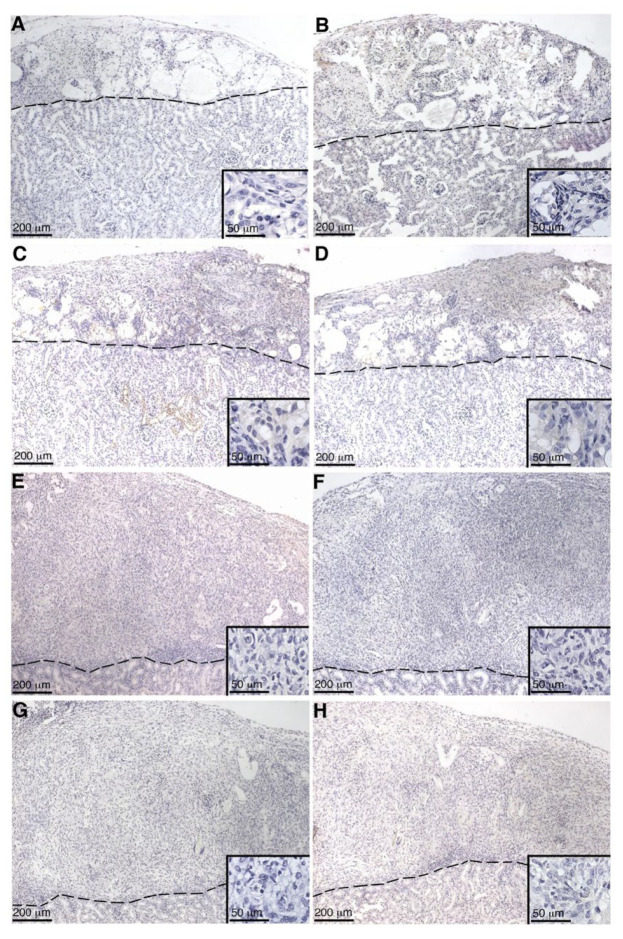
Immunohistochemical analysis of SC grafts for C4 and MAC deposition. Graft-bearing kidneys were collected at days 2 (**A**,**B**), 5 (**C**,**D**), 14 (**E**,**F**), and 20 (**G**,**H**) days post transplantation, and tissue sections were immunostained for C4 (brown; **A**,**C**,**E**,**G**) or MAC (brown; **B**,**D**,**F**,**H**). Insets are higher magnifications. A dotted line separates the graft (above the line) from the kidney (below the line). Sections were counterstained with hematoxylin (blue).

**Figure 8 ijms-23-12760-f008:**
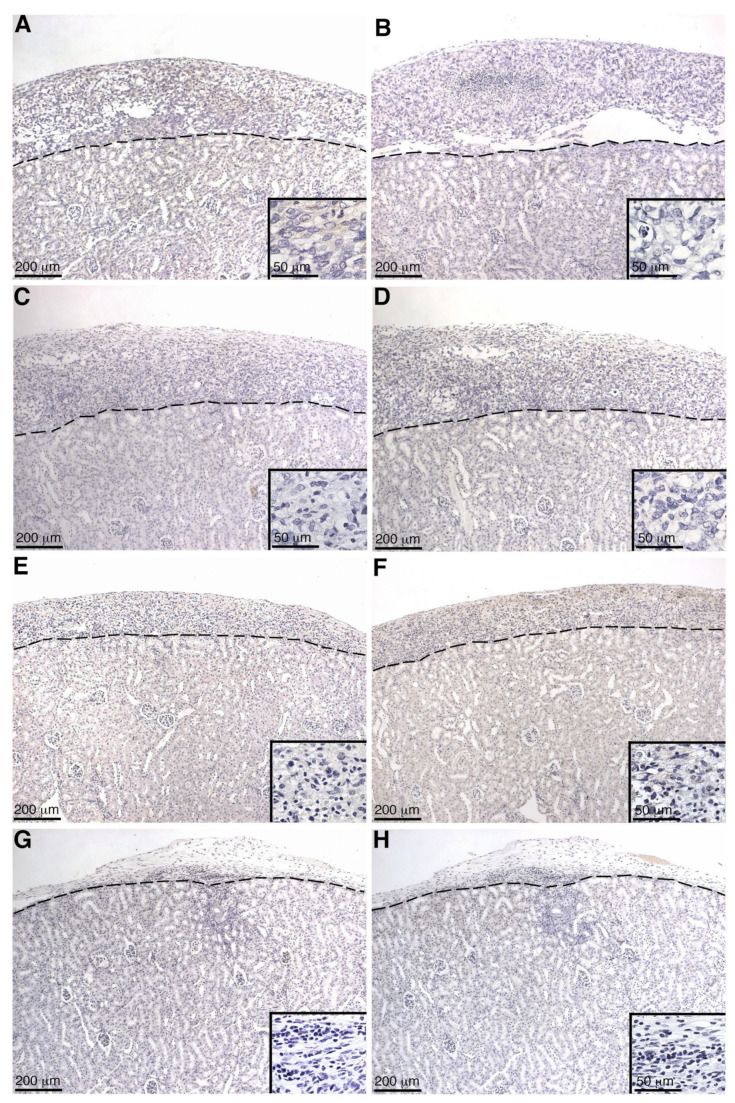
Immunohistochemical analysis of MSC-1 grafts for C4 and MAC deposition. Graft-bearing kidneys were collected at days 2 (**A**,**B**), 5 (**C**,**D**), 14 (**E**,**F**), and 20 (**G**,**H**) days post transplantation and tissue sections were immunostained for C4 (brown; **A**,**C**,**E**,**G**) or MAC (brown; **B**,**D**,**F**,**H**). Insets are higher magnifications. A dotted line separates the graft (above the line) from the kidney (below the line). Sections were counterstained with hematoxylin (blue).

**Figure 9 ijms-23-12760-f009:**
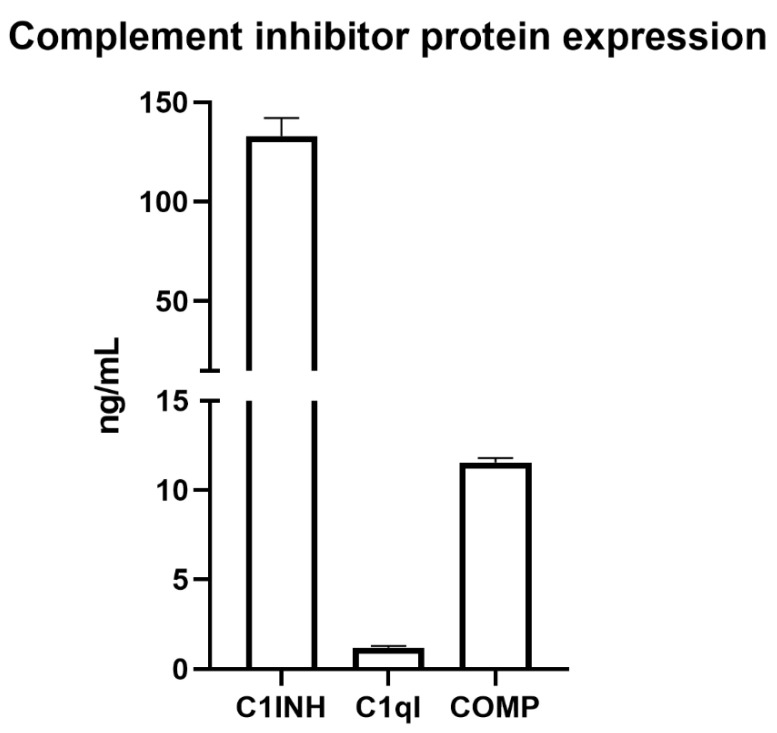
Mouse Sertoli cells express and secrete protein for complement inhibitors. Mouse SCCM was analyzed for protein levels of the complement inhibitors C1INH, C1qI, and COMP. C1INH was detected at 133.065 ± 9.001 ng/mL, C1qI was detected at 1.205 ± 0.092 ng/mL, and COMP was detected at 11.560 ± 0.231 ng/mL.

**Table 1 ijms-23-12760-t001:** Deposition of IgM and IgG on SC and MSC-1 grafts.

Days Post-	IgM-Positive Cells		IgG-Positive Cells	
Transplantation	SC	MSC-1	SC	MSC-1
1	0	0	0	0
2	0	0	0	0
5	0	0	0	0
8	0	0	0	0
11	0	0	0	0
14	0	0–15	0	0–4
20	0–8	0–6	0–10	0–4

Three independent experiments were performed with *n* = 3 for each graft type at each timepoint. The number of IgM and IgG positive cells was determined by counting the positive cells per each timepoint.

**Table 2 ijms-23-12760-t002:** Deposition of C4, C3, and MAC on SC and MSC-1 grafts.

Days Post-	C4-Positive Cells		C3-Positive Cells		MAC-Positive Cells	
Transplantation	SC	MSC-1	SC	MSC-1	SC	MSC-1
1	0	0	0–1	0	0–1	0
2	0	0	0	0	0–1	0–2
5	0	0	0–1	0	0	0
8	0	0	0	0	0	0
11	0	0	0	0	0–2	0–1
14	0	0	0	0	0–1	0
20	0–2	0	0	0	0–1	0

Three independent experiments were performed with *n* = 3 for each graft type at each timepoint. The number of C3-, C4-, and MAC-positive cells was determined by counting the positive cells per section for each timepoint.

**Table 3 ijms-23-12760-t003:** Complement inhibitor expression by mouse SCs.

Complement Inhibitor	SC	MSC-1	Fold Change	*p*-Value
C1INH *	1492	<50	64	0.01
C1qI *	2795	5811	−2	0.002
PTX3	257	1094	−3	0.07
COMP *	525	<50	13	0.001
SUSD4	143	107	0	0.10
Crry	738	632	0	0.37
Factor I *	72	<50	1	0.04
Factor H	67	<50	6	0.35
CD46	54	63	0	0.11
CD55 *	380	<50	28	0.002
CD59 *	218	2082	−9	0.0001
Clusterin *	20,218	2065	9	0.01
CPB *	65	<50	2	0.0001
CPN *	161	<50	3	0.0001

Expression level was measured by fluorescence intensity, and fold change was determined by comparing SCs to MSC-1 cells. Asterisks (*) denote complement inhibitors with significantly differentially expressed genes with fold change >±1 and *p* < 0.05 when comparing SC to MSC-1 cell expression.

## Data Availability

Microarray data referred to in this study is contained in Doyle et. al., 2012 [10].

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
