# Peer review of "Mouse Sertoli Cells Inhibit Humoral-Based Immunity"

_ijms, 2022, doi:10.3390/ijms232112760_

Round 1

Reviewer 1 Report

The research proposal is interesting on a topic of great interest, in a field of research that has recently made many advances in basic science and clinical research; Unfortunately this manuscript does not include the methods section which is extremely important as it provides the essential information to evaluate the validity of the results and conclusions of the study. The little information authors included about the study procedures is in the introduction and was presented incompletely.

For this reason, an objective evaluation of this work cannot be made, at least in this version.

Author Response

Response to Reviewer 1 Comments

Point 1: The research proposal is interesting on a topic of great interest, in a field of research that has recently made many advances in basic science and clinical research; Unfortunately this manuscript does not include the methods section which is extremely important as it provides the essential information to evaluate the validity of the results and conclusions of the study. The little information authors included about the study procedures is in the introduction and was presented incompletely.

For this reason, an objective evaluation of this work cannot be made, at least in this version.

Thank you for your positive comments and for agreeing to review our manuscript. We actually did include a methods section. It is after the Discussion, in section 4 (lines 424-530), per the journal formatting guidelines.

Reviewer 2 Report

In this article, Washburn et al. assess the activation of the complement system in response to the allograft of Sertoli cells. They observed that the Sertoli cells showed substantial protection against humoral immunity. The microarray experiment with the mouse Sertoli cells showed enhanced expression of complement inhibitory proteins (CIPs). They conclude by suggesting that since Sertoli cells exhibit protection from complement-mediated tissue rejection, it could be a novel strategy in diabetes for enhancing the success of islet grafts.

Overall, the study seems to highlight the protective mechanism exhibited by Sertoli cells’ graft in treating diabetes. The study is in line with a previous study (Fallarino et al., 2009) and one from the same group (Kaur et al., 2018). Although the findings are interesting and certainly pave the path for exciting avenues, at the current stage, several factors limit the enthusiasm in this study:

1. The study gives a substantial background, introduction, and future directions on diabetes treatment. Even the title talks about this application. However, none of the experiments or animal models provide insight into diabetes pathogenesis and treatment/prevention. Hence, it seems misleading. Perhaps the authors can consider eliminating diabetes treatment from most sections and include it in their discussion or conclusion section. If they wish to stick to the diabetes treatments, they must include experiments in the respective models.

2. Another major pitfall is the lack of a robust mechanism. It is still unclear how the Sertoli cells prevent complement activation. Although the authors demonstrate with microarray studies that the Sertoli cells have enhanced gene expression of CIPs, whether this carries over to the translational level remains to be elucidated. Hence, experiments demonstrating that the CIPs are produced by the SCs, which is the mechanism of protection against rejection, are warranted.

3. The authors have performed IHCs to show antibody binding. However, the images are unclear. Furthermore, the authors need to explain the figures better. Just pointing out the colors of the proteins is insufficient for clear understanding. They ought to describe the differences in changing conditions. Performing immunofluorescence would be a better alternative to demonstrate clear antibody binding.

4. The rationale for a lot of studying SV40 antigen as a marker of cell survival is unclear. Why haven’t they looked at classical markers, including cleaved caspases and/or TUNEL.

5. The authors observed Wilm’s tumor-positive pSCs in grafted animals. Does it mean that such grafting can promote tumors? It diminishes the objective of their study.

6. A few conclusions are unclear:

a. “By day 20 post-transplantation, MSC-1 cells were completely rejected (Figure 3B) as no large T-antigen positive cells were detected at this time point. (Lines 112-114).” – Isn’t this against the hypothesis? The Sertoli cells are supposed to be stably grafted, right?

b. “Overall, this indicates both pSC and MSC-1 cells inhibit antibody and complement-mediated killing in vitro while the response in vivo is more complicated (Lines 115-116)”. The basis of this conclusion is unclear as it lacks robust evidence.

c. “Mice transplanted with pSC showed no significant change in IgM response throughout the study compared to naïve mice (Figure 4, white bars). However, a significant increase in IgM production on days 5 and 8 post-transplantation was detected in mice transplanted with MSC-1 cells (Figure 4, gray bars). Furthermore, IgM  levels in mice transplanted with pSC grafts were significantly lower than those transplanted with MSC-1 cells on day 8 post-transplantation yet significantly higher at day 14 post-transplantation (Lines 134-139)” . 

The authors need to explain why they see these findings. Also, the authors need to explain the rationale behind a distorted timeline of assessment.

d. “As alloantibody binding to the surface of transplanted cells is implicated in activating complement in transplant rejection, pSC or MSC-1 cell graft tissue was stained for IgM or IgG. In pSC grafts, neither IgM or IgG were detected at days 1-14 post-transplantation (Table 1 and Figure 5A-F), whereas at day 20 post-transplantation, both IgM and IgG deposition was detected in pSC grafts (Table 1, Figure 5G and H). In MSC-1 cell grafts, IgM and IgG deposition was not detected until day 14 and 20 post-transplantation (Table 1, Figure 6A-D). Higher IgM deposition was detected as compared to IgG (Lines 143-149)”.

Here the authors need to address why do they observe a discrepancy in the detection of antibodies on the Sertoli cells in this figure versus observations in the previous figure.

7. The authors seemed to have incorrectly labeled procedures as ‘allografts’ instead of ‘xenografts’ (For e.g. grafting porcine cells in mice constitutes xenografts). It needs to be corrected.

8. In the materials and methods section, there are some instances where the authors mention that the technique has been performed as previously described. It is highly recommended that the authors briefly describe the procedure again. This limits the efforts of the reader to track back the manuscripts for one method. For example, in line 443, the authors mention, “Microarray processing and data analyses were performed by Doyle et al. 2012”.

9. The authors also need to work on the discussion section thoroughly. It needs to be more scholarly rather than a mere re-iteration of results.

Author Response

Response to Reviewer 2 Comments

In this article, Washburn et al. assess the activation of the complement system in response to the allograft of Sertoli cells. They observed that the Sertoli cells showed substantial protection against humoral immunity. The microarray experiment with the mouse Sertoli cells showed enhanced expression of complement inhibitory proteins (CIPs). They conclude by suggesting that since Sertoli cells exhibit protection from complement-mediated tissue rejection, it could be a novel strategy in diabetes for enhancing the success of islet grafts.

Overall, the study seems to highlight the protective mechanism exhibited by Sertoli cells’ graft in treating diabetes. The study is in line with a previous study (Fallarino et al., 2009) and one from the same group (Kaur et al., 2018). Although the findings are interesting and certainly pave the path for exciting avenues, at the current stage, several factors limit the enthusiasm in this study:

  1. The study gives a substantial background, introduction, and future directions on diabetes treatment. Even the title talks about this application. However, none of the experiments or animal models provide insight into diabetes pathogenesis and treatment/prevention. Hence, it seems misleading. Perhaps the authors can consider eliminating diabetes treatment from most sections and include it in their discussion or conclusion section. If they wish to stick to the diabetes treatments, they must include experiments in the respective models.

We agree with the reviewer’s comment. Previously, we had planned to include data on insulin-producing Sertoli cells in diabetic mice to fit with the journal topic of metabolic disorders. However, we decided to remove this data since we felt it needs further analyses. We have changed the focus and significance of this paper to transplantation.

  1. Another major pitfall is the lack of a robust mechanism. It is still unclear how the Sertoli cells prevent complement activation. Although the authors demonstrate with microarray studies that the Sertoli cells have enhanced gene expression of CIPs, whether this carries over to the translational level remains to be elucidated. Hence, experiments demonstrating that the CIPs are produced by the SCs, which is the mechanism of protection against rejection, are warranted.

The purpose of this study was to assess potential antibody and complement-mediated killing of primary mouse Sertoli cells and a mouse Sertoli cell line (MSC-1, which lacks some of the protective qualities of primary Sertoli cells) in vitro against human serum and in vivo in an allograft model. We suggest that a possible mechanism of protection against complement in both these cell grafts could be expression of CIPs, which is why we analyzed a previous data set for this expression. We agree that this has not been established as a robust mechanism, and we intend to further investigate the importance of CIP expression in SC survival in future studies.

  1. The authors have performed IHCs to show antibody binding. However, the images are unclear. Furthermore, the authors need to explain the figures better. Just pointing out the colors of the proteins is insufficient for clear understanding. They ought to describe the differences in changing conditions. Performing immunofluorescence would be a better alternative to demonstrate clear antibody binding.

We have added arrows to point out the positively stained cells and have updated the figure legends to be clearer. We have also added more description in the text.

  1. The rationale for a lot of studying SV40 antigen as a marker of cell survival is unclear. Why haven’t they looked at classical markers, including cleaved caspases and/or TUNEL.

We have clarified our reasoning for studying the SV40 antigen as a marker of MSC-1 cell survival. We have previously completed TUNEL analyses to confirm these results and now refer to the TUNEL findings in the text and referenced paper (lines 130-132 and 443-448).

  1. The authors observed Wilm’s tumor-positive pSCs in grafted animals. Does it mean that such grafting can promote tumors? It diminishes the objective of their study.

Wilm’s tumor is a gene expressed by pSC which is important in mouse spermatogenesis. Because SC normally express it, it is used as a marker for SC survival. We have clarified this in the text (lines 126-130). Despite the expression of WT1, SC are considered terminally differentiated and very rarely form tumors. We have previously analyzed SC grafts for proliferation (PCNA and BrdU labeling) and found that the SC stop proliferating within the first 10 to 20 days after transplantation and are not forming tumors. This has been added to the discussion (lines 303-307).

  1. A few conclusions are unclear:
  2. “By day 20 post-transplantation, MSC-1 cells were completely rejected (Figure 3B) as no large T-antigen positive cells were detected at this time point. (Lines 112-114).” – Isn’t this against the hypothesis? The Sertoli cells are supposed to be stably grafted, right?

Primary, freshly isolated SCs survive long-term after being transplanted as allografts or xenografts. MSC-1 cells are a SC line that lack some of the immunoregulatory properties of primary Sertoli cells and we have found reject when transplanted as allografts within 20 days post-transplantation. We used them as controls for comparison as we think the differences provide important information on SC immune regulation. We have clarified this in the text and referenced another paper that also describes the rejection of the MSC-1 cells (lines 109-110, 133-140, and 293-302).

  1. “Overall, this indicates both pSC and MSC-1 cells inhibit antibody and complement-mediated killing in vitro while the response in vivo is more complicated (Lines 115-116)”. The basis of this conclusion is unclear as it lacks robust evidence.

We found that SCs survive xenogeneic human antibody and complement-mediated killing in vitro, allogeneic complement-mediated destruction without immune suppressants in mice, and that SCs express most of the known complement inhibitors. However, MSC-1 cells also expressed CIPs and survived the in vitro human serum assay but were rejected after allograft transplantation. We believe this indicates that SC survival of acute rejection is more complicated than solely inhibiting complement-mediated cytolysis, and that other mechanisms may be at play including regulation of cellular immunity. We clarified this statement (lines 109-110, 133-140, 293-302, and 535-539).

  1. “Mice transplanted with pSC showed no significant change in IgM response throughout the study compared to naïve mice (Figure 4, white bars). However, a significant increase in IgM production on days 5 and 8 post-transplantation was detected in mice transplanted with MSC-1 cells (Figure 4, gray bars). Furthermore, IgM levels in mice transplanted with pSC grafts were significantly lower than those transplanted with MSC-1 cells on day 8 post-transplantation yet significantly higher at day 14 post-transplantation (Lines 134-139)”.  The authors need to explain why they see these findings. Also, the authors need to explain the rationale behind a distorted timeline of assessment.

We added a section in the discussion to explain these findings further (lines 308-322). The timepoints chosen for analysis were selected to cover the immune response during the acute phase of graft rejection.

  1. “As alloantibody binding to the surface of transplanted cells is implicated in activating complement in transplant rejection, pSC or MSC-1 cell graft tissue was stained for IgM or IgG. In pSC grafts, neither IgM or IgG were detected at days 1-14 post-transplantation (Table 1 and Figure 5A-F), whereas at day 20 post-transplantation, both IgM and IgG deposition was detected in pSC grafts (Table 1, Figure 5G and H). In MSC-1 cell grafts, IgM and IgG deposition was not detected until day 14 and 20 post-transplantation (Table 1, Figure 6A-D). Higher IgM deposition was detected as compared to IgG (Lines 143-149)”.

Here the authors need to address why do they observe a discrepancy in the detection of antibodies on the Sertoli cells in this figure versus observations in the previous figure.

We addressed this in lines 323-331.

  1. The authors seemed to have incorrectly labeled procedures as ‘allografts’ instead of ‘xenografts’ (For e.g. grafting porcine cells in mice constitutes xenografts). It needs to be corrected.

We clarified this throughout the text since we used an in vitro xenograft model and an in vivo allograft model (lines 107-109, 125-126, and 455-459). We also clarified this in our discussion when we discussed previous xenograft studies (lines 273-274, 299-300, 354-357, and 532-538).

  1. In the materials and methods section, there are some instances where the authors mention that the technique has been performed as previously described. It is highly recommended that the authors briefly describe the procedure again. This limits the efforts of the reader to track back the manuscripts for one method. For example, in line 443, the authors mention, “Microarray processing and data analyses were performed by Doyle et al. 2012”.

We corrected this and included brief descriptions in lines 436-452, 459-468, and 512-524.

  1. The authors also need to work on the discussion section thoroughly. It needs to be more scholarly rather than a mere re-iteration of results.

The discussion has been modified as requested.

Reviewer 3 Report

In this article, Washburn et al. assess the activation of the complement system in response to the allograft of Sertoli cells. They observed that the Sertoli cells showed substantial protection against humoral immunity. The microarray experiment with the mouse Sertoli cells showed enhanced expression of complement inhibitory proteins (CIPs). They conclude by suggesting that since Sertoli cells exhibit protection from complement-mediated tissue rejection, it could be a novel strategy in diabetes for enhancing the success of islet grafts.

Overall, the study seems to highlight the protective mechanism exhibited by Sertoli cells’ graft in treating diabetes. The study is in line with a previous study (Fallarino et al., 2009) and one from the same group (Kaur et al., 2018). Although the findings are interesting and certainly pave the path for exciting avenues, at the current stage, several factors limit the enthusiasm in this study:

1. The study gives a substantial background, introduction, and future directions on diabetes treatment. Even the title talks about this application. However, none of the experiments or animal models provide insight into diabetes pathogenesis and treatment/prevention. Hence, it seems misleading. Perhaps the authors can consider eliminating diabetes treatment from most sections and include it in their discussion or conclusion section. If they wish to stick to the diabetes treatments, they must include experiments in the respective models.

2. Another major pitfall is the lack of a robust mechanism. It is still unclear how the Sertoli cells prevent complement activation. Although the authors demonstrate with microarray studies that the Sertoli cells have enhanced gene expression of CIPs, whether this carries over to the translational level remains to be elucidated. Hence, experiments demonstrating that the CIPs are produced by the SCs, which is the mechanism of protection against rejection, are warranted.

3. The authors have performed IHCs to show antibody binding. However, the images are unclear. Furthermore, the authors need to explain the figures better. Just pointing out the colors of the proteins is insufficient for clear understanding. They ought to describe the differences in changing conditions. Performing immunofluorescence would be a better alternative to demonstrate clear antibody binding.

4. The rationale for a lot of studying SV40 antigen as a marker of cell survival is unclear. Why haven’t they looked at classical markers, including cleaved caspases and/or TUNEL.

5. The authors observed Wilm’s tumor-positive pSCs in grafted animals. Does it mean that such grafting can promote tumors? It diminishes the objective of their study.

6. A few conclusions are unclear:

a. “By day 20 post-transplantation, MSC-1 cells were completely rejected (Figure 3B) as no large T-antigen positive cells were detected at this time point. (Lines 112-114).” – Isn’t this against the hypothesis? The Sertoli cells are supposed to be stably grafted, right?

b. “Overall, this indicates both pSC and MSC-1 cells inhibit antibody and complement-mediated killing in vitro while the response in vivo is more complicated (Lines 115-116)”. The basis of this conclusion is unclear as it lacks robust evidence.

c. “Mice transplanted with pSC showed no significant change in IgM response throughout the study compared to naïve mice (Figure 4, white bars). However, a significant increase in IgM production on days 5 and 8 post-transplantation was detected in mice transplanted with MSC-1 cells (Figure 4, gray bars). Furthermore, IgM  levels in mice transplanted with pSC grafts were significantly lower than those transplanted with MSC-1 cells on day 8 post-transplantation yet significantly higher at day 14 post-transplantation (Lines 134-139)” . 

The authors need to explain why they see these findings. Also, the authors need to explain the rationale behind a distorted timeline of assessment.

d. “As alloantibody binding to the surface of transplanted cells is implicated in activating complement in transplant rejection, pSC or MSC-1 cell graft tissue was stained for IgM or IgG. In pSC grafts, neither IgM or IgG were detected at days 1-14 post-transplantation (Table 1 and Figure 5A-F), whereas at day 20 post-transplantation, both IgM and IgG deposition was detected in pSC grafts (Table 1, Figure 5G and H). In MSC-1 cell grafts, IgM and IgG deposition was not detected until day 14 and 20 post-transplantation (Table 1, Figure 6A-D). Higher IgM deposition was detected as compared to IgG (Lines 143-149)”.

Here the authors need to address why do they observe a discrepancy in the detection of antibodies on the Sertoli cells in this figure versus observations in the previous figure.

7. The authors seemed to have incorrectly labeled procedures as ‘allografts’ instead of ‘xenografts’ (For e.g. grafting porcine cells in mice constitutes xenografts). It needs to be corrected.

8. In the materials and methods section, there are some instances where the authors mention that the technique has been performed as previously described. It is highly recommended that the authors briefly describe the procedure again. This limits the efforts of the reader to track back the manuscripts for one method. For example, in line 443, the authors mention, “Microarray processing and data analyses were performed by Doyle et al. 2012”.

9. The authors also need to work on the discussion section thoroughly. It needs to be more scholarly rather than a mere re-iteration of results.

Author Response

Response to Reviewer 3 Comments

In this article, Washburn et al. assess the activation of the complement system in response to the allograft of Sertoli cells. They observed that the Sertoli cells showed substantial protection against humoral immunity. The microarray experiment with the mouse Sertoli cells showed enhanced expression of complement inhibitory proteins (CIPs). They conclude by suggesting that since Sertoli cells exhibit protection from complement-mediated tissue rejection, it could be a novel strategy in diabetes for enhancing the success of islet grafts.

Overall, the study seems to highlight the protective mechanism exhibited by Sertoli cells’ graft in treating diabetes. The study is in line with a previous study (Fallarino et al., 2009) and one from the same group (Kaur et al., 2018). Although the findings are interesting and certainly pave the path for exciting avenues, at the current stage, several factors limit the enthusiasm in this study:

  1. The study gives a substantial background, introduction, and future directions on diabetes treatment. Even the title talks about this application. However, none of the experiments or animal models provide insight into diabetes pathogenesis and treatment/prevention. Hence, it seems misleading. Perhaps the authors can consider eliminating diabetes treatment from most sections and include it in their discussion or conclusion section. If they wish to stick to the diabetes treatments, they must include experiments in the respective models.

We agree with the reviewer’s comment. Previously, we had planned to include data on insulin-producing Sertoli cells in diabetic mice to fit with the journal topic of metabolic disorders. However, we decided to remove this data since we felt it needs further analyses. We have changed the focus and significance of this paper to transplantation.

  1. Another major pitfall is the lack of a robust mechanism. It is still unclear how the Sertoli cells prevent complement activation. Although the authors demonstrate with microarray studies that the Sertoli cells have enhanced gene expression of CIPs, whether this carries over to the translational level remains to be elucidated. Hence, experiments demonstrating that the CIPs are produced by the SCs, which is the mechanism of protection against rejection, are warranted.

The purpose of this study was to assess potential antibody and complement-mediated killing of primary mouse Sertoli cells and a mouse Sertoli cell line (MSC-1, which lacks some of the protective qualities of primary Sertoli cells) in vitro against human serum and in vivo in an allograft model. We suggest that a possible mechanism of protection against complement in both these cell grafts could be expression of CIPs, which is why we analyzed a previous data set for this expression. We agree that this has not been established as a robust mechanism, and we intend to further investigate the importance of CIP expression in SC survival in future studies.

  1. The authors have performed IHCs to show antibody binding. However, the images are unclear. Furthermore, the authors need to explain the figures better. Just pointing out the colors of the proteins is insufficient for clear understanding. They ought to describe the differences in changing conditions. Performing immunofluorescence would be a better alternative to demonstrate clear antibody binding.

We have added arrows to point out the positively stained cells and have updated the figure legends to be clearer. We have also added more description in the text.

  1. The rationale for a lot of studying SV40 antigen as a marker of cell survival is unclear. Why haven’t they looked at classical markers, including cleaved caspases and/or TUNEL.

We have clarified our reasoning for studying the SV40 antigen as a marker of MSC-1 cell survival. We have previously completed TUNEL analyses to confirm these results and now refer to the TUNEL findings in the text and referenced paper (lines 130-132 and 443-448).

  1. The authors observed Wilm’s tumor-positive pSCs in grafted animals. Does it mean that such grafting can promote tumors? It diminishes the objective of their study.

Wilm’s tumor is a gene expressed by pSC which is important in mouse spermatogenesis. Because SC normally express it, it is used as a marker for SC survival. We have clarified this in the text (lines 126-130). Despite the expression of WT1, SC are considered terminally differentiated and very rarely form tumors. We have previously analyzed SC grafts for proliferation (PCNA and BrdU labeling) and found that the SC stop proliferating within the first 10 to 20 days after transplantation and are not forming tumors. This has been added to the discussion (lines 303-307).

  1. A few conclusions are unclear:
  2. “By day 20 post-transplantation, MSC-1 cells were completely rejected (Figure 3B) as no large T-antigen positive cells were detected at this time point. (Lines 112-114).” – Isn’t this against the hypothesis? The Sertoli cells are supposed to be stably grafted, right?

Primary, freshly isolated SCs survive long-term after being transplanted as allografts or xenografts. MSC-1 cells are a SC line that lack some of the immunoregulatory properties of primary Sertoli cells and we have found reject when transplanted as allografts within 20 days post-transplantation. We used them as controls for comparison as we think the differences provide important information on SC immune regulation. We have clarified this in the text and referenced another paper that also describes the rejection of the MSC-1 cells (lines 109-110, 133-140, and 293-302).

  1. “Overall, this indicates both pSC and MSC-1 cells inhibit antibody and complement-mediated killing in vitro while the response in vivo is more complicated (Lines 115-116)”. The basis of this conclusion is unclear as it lacks robust evidence.

We found that SCs survive xenogeneic human antibody and complement-mediated killing in vitro, allogeneic complement-mediated destruction without immune suppressants in mice, and that SCs express most of the known complement inhibitors. However, MSC-1 cells also expressed CIPs and survived the in vitro human serum assay but were rejected after allograft transplantation. We believe this indicates that SC survival of acute rejection is more complicated than solely inhibiting complement-mediated cytolysis, and that other mechanisms may be at play including regulation of cellular immunity. We clarified this statement (lines 109-110, 133-140, 293-302, and 535-539).

  1. “Mice transplanted with pSC showed no significant change in IgM response throughout the study compared to naïve mice (Figure 4, white bars). However, a significant increase in IgM production on days 5 and 8 post-transplantation was detected in mice transplanted with MSC-1 cells (Figure 4, gray bars). Furthermore, IgM levels in mice transplanted with pSC grafts were significantly lower than those transplanted with MSC-1 cells on day 8 post-transplantation yet significantly higher at day 14 post-transplantation (Lines 134-139)”.  The authors need to explain why they see these findings. Also, the authors need to explain the rationale behind a distorted timeline of assessment.

We added a section in the discussion to explain these findings further (lines 308-322). The timepoints chosen for analysis were selected to cover the immune response during the acute phase of graft rejection.

  1. “As alloantibody binding to the surface of transplanted cells is implicated in activating complement in transplant rejection, pSC or MSC-1 cell graft tissue was stained for IgM or IgG. In pSC grafts, neither IgM or IgG were detected at days 1-14 post-transplantation (Table 1 and Figure 5A-F), whereas at day 20 post-transplantation, both IgM and IgG deposition was detected in pSC grafts (Table 1, Figure 5G and H). In MSC-1 cell grafts, IgM and IgG deposition was not detected until day 14 and 20 post-transplantation (Table 1, Figure 6A-D). Higher IgM deposition was detected as compared to IgG (Lines 143-149)”.

Here the authors need to address why do they observe a discrepancy in the detection of antibodies on the Sertoli cells in this figure versus observations in the previous figure.

We addressed this in lines 323-331.

  1. The authors seemed to have incorrectly labeled procedures as ‘allografts’ instead of ‘xenografts’ (For e.g. grafting porcine cells in mice constitutes xenografts). It needs to be corrected.

We clarified this throughout the text since we used an in vitro xenograft model and an in vivo allograft model (lines 107-109, 125-126, and 455-459). We also clarified this in our discussion when we discussed previous xenograft studies (lines 273-274, 299-300, 354-357, and 532-538).

  1. In the materials and methods section, there are some instances where the authors mention that the technique has been performed as previously described. It is highly recommended that the authors briefly describe the procedure again. This limits the efforts of the reader to track back the manuscripts for one method. For example, in line 443, the authors mention, “Microarray processing and data analyses were performed by Doyle et al. 2012”.

We corrected this and included brief descriptions in lines 436-452, 459-468, and 512-524.

  1. The authors also need to work on the discussion section thoroughly. It needs to be more scholarly rather than a mere re-iteration of results.

The discussion has been modified as requested.

Round 2

Reviewer 1 Report

 I would like to thank the authors, they have addressed previous  comments. They have presented an improved version of the manuscript.

Author Response

Thank you so much for reviewing our manuscript.

Reviewer 2 Report

The authors have done a great job in addressing most of my comments. The manuscript certainly looks better and flows well now.

The only experiment I would still encourage the authors to do is any protein analysis to validate their hypothesis regarding CIPs. Although the authors acknowledge that whatever they have provided doesn't constitute a robust mechanism and would focus on it in their future studies, they must leave a strong message from this current version. For that, some form of protein analysis (ELISA, FLOW, or Western blot) would be necessary.

Reviewer 3 Report

The authors have done a great job in addressing most of my comments. The manuscript certainly looks better and flows well now.

The only experiment I would still encourage the authors to do is any protein analysis to validate their hypothesis regarding CIPs. Although the authors acknowledge that whatever they have provided doesn't constitute a robust mechanism and would focus on it in their future studies, they must leave a strong message from this current version. For that, some form of protein analysis (ELISA, FLOW, or Western blot) would be necessary.
